# A Proton Battery Stack Real-Time Monitor with a Flexible Six-in-One Microsensor

**DOI:** 10.3390/membranes12080779

**Published:** 2022-08-13

**Authors:** Chi-Yuan Lee, Chia-Hung Chen, Yun-Hsiu Chien, Zhi-Yu Huang

**Affiliations:** 1Department of Mechanical Engineering, Yuan Ze Fuel Cell Center, Yuan Ze University, Taoyuan 32003, Taiwan; 2Homytech Global Co., Ltd., Taoyuan 33464, Taiwan

**Keywords:** proton battery stack, flexible six-in-one microsensor, real-time monitoring

## Abstract

A proton battery is a hybrid battery produced by combining a hydrogen fuel cell and a battery system in an attempt to obtain the advantages of both systems. As the battery life of a single proton battery is not good, the proton battery stack is developed by connecting in parallel, which can greatly improve the battery life of proton batteries. In order to obtain important information about the proton battery stack in real time, a flexible six-in-one microsensor is embedded in the proton battery stack. This study has successfully developed a health diagnostic tool for a proton battery stack using micro-electro-mechanical systems (MEMS) technology. This study also focused on the innovatively developed hydrogen microsensor, and integrated the voltage, current, temperature, humidity, and flow microsensors, as previously developed by our laboratory, to complete the flexible six-in-one microsensor. Six important internal physical parameters were simultaneously measured during the entire operation of the proton battery stack. It also established a complete database and monitor system in real time to detect the internal health status of the proton cell stack and observe if there were problems, such as water accumulation, aging, or failure, in order to understand the changes and effects of the various physical quantities of long-term operation. The study found that the proton batteries exhibited significant differences in the hydrogen absorb rates and hydrogen release rates. The ceramic circuit board used in the original sensor is replaced by a flexible board to improve problems such as peeling and breaking.

## 1. Introduction

Net-zero carbon emissions have become a major global trend, and as of September 2021, 55 countries have committed to achieving their national net-zero carbon emissions targets [1]. It is important to use fuel cell technology to reduce carbon emissions, and because fuel cell technology has become increasingly mature, it has begun to be widely used in various fields. However, there are still problems that need to be solved and overcome, such as the requirements of small size and lightweight fuel cells and the easy storage of hydrogen gas. In 2018, Heidari et al. [2] led the development of a proton battery that combines a hydrogen storage tank and a fuel cell without another hydrogen storage tank, as the fuel cell has the function of hydrogen storage, which can reduce the risk of hydrogen leakage while avoiding the required space and cost of hydrogen storage tanks. In 2019, Kapoor et al. [3] used multi-wall carbon nanotubes for hydrogen storage and confirmed their experimental results. When using porous materials for hydrogen storage, the charging current must be kept at a low value to avoid the generation of hydrogen. The main fuel cell application concept is to pre-store hydrogen in hydrogen storage tanks, and then, install the hydrogen storage tanks in vehicles, such as cars and generators, for use. As first proposed by Andrews et al. [4], most hydrogen storage tanks use metal hydride technology to store hydrogen, which uses the metal hydride–nitride composite electrode of homogeneous regenerative fuel cells. The hydrogen stored in the electrode was measured to be 0.6 wt% in the charge mode, and while the amount of hydrogen emitted was detectable, at only about 0.01 wt%, it posed a safety risk [5]. Zhou et al. [6] studied the effects of stack end plate material, the number of single cells, and stack compressibility on stack performance. Among them, the gas diffusion layer (GDL) has the characteristics of nonlinear compression, which leads to poor pressure uniformity when the pressure is increased, the more the stacks, the higher the compression ratio and the smaller the gap. Arifin et al. [7] used rice husk-derived graphene to prepare activated carbon combined with a zeolitic imidazolate framework (ZIF-8), where the Brunauer–Emmet–Teller (BET) surface area ratio was three times that of pristine carbon materials. Bosch et al. [8] investigated increasing the surface area and improving materials. By understanding the hydrogen storage properties of carbon nanostructures, the feasibility of high energy density, high hydrogen storage capacity, and rechargeable proton batteries can be increased. As the proton battery is a very new research field with some immature prototypes [9,10,11], it is worth further research.

Chakik et al. [12] observed data regarding proton battery operation and found that the amount of hydrogen gas produced increased when a current was continuously applied in a range of current strengths (0.2 A to 0.5 A) at a constant voltage. Chahartaghi et al. [13] proposed a novel concept of a combined cooling heating and power (CCHP) system using a proton exchange membrane fuel cell combined with an absorption cooler and thermal storage tank and investigated the effect of the fuel cell operating current density of the energy efficiency of the fuel cell and CCHP. The results showed that when the current density was increased, the fuel cell efficiency decreased. Whereas, the CCHP efficiency initially increased to a maximum value, then, decreased, which is because the irreversible concentration loss increased sharply with the increasing current density. Ou et al. [14] suggested that the performance of fuel cells depends on changes in temperature, and they tested the effect of temperature on fuel cell performance with a variety of different temperatures to confirm the effect of temperature on performance. Sun et al. [15] used micro-electro-mechanical systems (MEMS) to fabricate a calorimetric flow sensor, and their research showed that the higher the thickness of the contaminant on the sensor, the greater its effect on the sensing accuracy of higher air. Kaya et al. [16] conducted experiments to study the effect of water flow and magnetic flux density on the performance of proton exchange membrane water electrolysis (PEMWE). Regarding the differences in different flows, a low magnetic flux produces the greatest current difference in low flow rates, while a high magnetic flux produces the greatest current difference in high flow rates. Kim et al. [17] conducted a parametric study on the performance of proton exchange membrane fuel cells (PEMFC) under dry conditions and found that when the relative humidity of the incoming gas was 25%, the membrane in the inlet area was dehydrated.

Kandidayeni et al. [18] mentioned that PEMFC is a multiphysics system and there is an interdependence between its power delivery and operating conditions. Flow rate and pressure inhomogeneity will accelerate the aging of membrane electrode assembly (MEA) membrane materials and lead to serious degradation of fuel cell performance. Therefore, it is very important to record these physical data with microsensors and analyze them. Previously, this team used flexible five-in-one microsensors for the real-time monitoring of proton batteries [19], while this research further incorporated a hydrogen microsensor to gain a clearer view of the interior of the proton battery stack. Since then, Cirrone et al. [20] performed a critical analysis of the hazards and associated risks for hydrogen-driven vehicles and transport through tunnels or similar confined spaces. Hydrogen leak will cause risk. Wang et al. [21] found that MoS_2_ has a unique thin-layer flower-like structure, which can be combined with SnO_2_ to greatly improve the sensitivity of a gas sensor, meaning that its sensing response and recovery speed are much higher than that of the original material. Abinaya et al. [22] used a DC magnetron sputtering technique to fabricate a hydrogen sensor with SnO_2_ thin films and confirmed that SnO_2_ will affect the sensing behavior of the sensor, as well as the effect of cathode power on sensitivity, response, and recovery time.

## 2. Sensing Principle of Flexible Six-in-One Microsensor

The six-in-one microsensor (voltage, current, temperature, flow, humidity, hydrogen) in this study was fabricated using MEMS technology. Polyimide (PI) is used as the base material, which has the advantages of wide application temperature, chemical corrosion resistance, and high strength. It is also suitable for use inside the proton cell stack. The adhesion layer at the hydrogen end immersed in sulfuric acid was chosen from Ti to avoid corrosion of the circuit and Cr for the rest of the adhesion layer. The process was selected the surface micromachining of MEMS technology [23].

### 2.1. Principle of Hydrogen Microsensor

The hydrogen microsensor of this study uses a semi-conductor-type gas-sensitive membrane sensor. The principle is to use the reductive gas and oxygen on the surface of the gas-sensitive membrane. The gas-sensitive membrane chosen for this experiment was tin dioxide (SnO_2_). When oxygen (O_2_) is adsorbed on the surface of the gas-sensitive membrane, the electrons inside the material are easily carried away to form oxygen ions (O-). Additionally, the resistance increases due to the reduction of electrons inside the gas-sensitive membrane. When the sensor is exposed to an environment filled with reductive gases (CO, H_2_, etc.), the reductive gases in the environment will react with the oxygen ions adsorbed on the surface of the gas-sensitive membrane. This causes the electrons to return to the inside of the gas-sensitive membrane, thus causing the resistance to drop. The reaction formula is shown in Equation (1).
H_2_ + ½ O_2_ → H_2_O(1)

### 2.2. Principle of Temperature Microsensor

The resistance temperature detector (RTD) is used in this study. The temperature sensing principle is based on the change in resistance caused by the change in temperature of the metal. Au is used as a temperature-sensitive resistive material because of its stable chemical properties, simple manufacturing process, and high linearity. The sensing principle is based on the following theory and Equations (2)–(5) mentioned in the book by Jewett and Serway [24]. In a limited temperature range, when the conductor resistivity varies slightly with temperature, it is expressed according to Equation (2).
*ρ* = *ρ*_0_ [1 + *α* (*T* − *T*_0_)](2)
where *ρ* is the resistivity at a certain temperature (T). *ρ*_0_ is the resistivity at a reference temperature (*T*_0_, generally 20 °C) and *α* is the temperature coefficient of resistance (TCR). Then, the Equation (2) can be expressed as (3):(3)$$\alpha =\frac{1}{\rho_{0}}\frac{d\rho }{dT}$$
where *α* is the resistance value and *dρ*(*dρ* = *ρ −*
$\rho_{0}$) is the rate of change of resistance between *dT*(*dT* = *T* − *T*_0_).

According to Equation (4) of general metal long and straight conductor, the resistance is proportional to the resistivity, that is, Equation (5) of resistance change.
(4)$$R=\rho \frac{L}{A}$$
(5)$$R_{t}=R_{0}\left(1+\alpha_{1}\Delta T\right)$$
where *R*_0_ is the resistance at *T*_0_, the temperature coefficient *α*_1_ is the sensitivity of the sensor, (5) can be rewritten as (6).
(6)$$\alpha_{1}=\frac{R_{t}-R_{0}}{R_{0}(\Delta T)}$$

### 2.3. Principle of Humidity Microsensor

Humidity microsensor using a resistive humidity microsensor. The non-conductive LTC 9305(Fujifilm Durimide^®^ LTC 9305) is used as the humidity sensing material. When the volume of the material increases with the amount of moisture absorbed, the resistance of the adhering circuit also increases.

### 2.4. Principle of Flow Microsensor

The measurement method used in this study is a hot-wire flow microsensor. The power supply is used to provide a stable voltage to generate a certain amount of heat at the sensing end. When a fluid passes through a heat source, it carries the heat away and causes a change in temperature, and the change in temperature causes a change in resistance. According to Ohm’s law, the current will change when the resistance is changed under the condition of constant voltage. In other words, thermal flow microsensor is a microsensor designed to positively correlate the heat energy dissipation rate of the hot wire with the fluid flow rate.

According to King’s law, the relationship between the rate of heat decay and the fluid flow rate is as shown in Equation (7) [25].
*Q* = *I*^2^ × *R* = *I* × *V* = (*A* + *B* × *U*(*n*)) (*Ts* − *To*)(7)
where *Q* is the electrical power supplied by the external power source; *U* is the flow rate of the fluid; n is the correlation coefficient between the heat *Q* and the flow velocity *U*, which is experimentally known to be about 0.5; *Ts* is the hot wire temperature; *To* is the temperature of the inlet fluid; *A* is a constant, When the flow rate is always zero, the heat coefficient transmitted by the heater; *B* is a constant, when the flow rate is not zero, the heat coefficient of the fluid and the heater is received; therefore, the Equation (7) can be rewritten as Equation (8)
*Q* = (*A* + *B* × *U*(0.5)) Δ*T*(8)

### 2.5. Principle of Voltage and Current Microsensor

Voltage and current microsensors use miniaturized metal probes. To ensure that the values are not affected by the remaining parts, the LTC 9320 (Fujifilm Durimide^®^ LTC 9320, MicroChemicals, Hsinchu, Taiwan) is used to cover the part of the lead wire that extends to the middle, as shown in the Figure 1.

## 3. Process Development of Flexible Six-in-One Microsensor

Using MEMS technology, this research successfully developed a sensing structure for six physical quantities of hydrogen, including temperature, voltage, current, flow, and humidity. The flowchart is shown in Figure 2.

(a)The PI film was first cleaned with an organic solution of acetone and methanol and used the ultrasonic shaker to shake and wash for three minutes. In order to remove surface dust and residual grease and increase the adhesion of the membrane metal, and the residual methanol was removed with deionized water and baked on a hot plate.(b)Then, an electron beam evaporator (EBS-500, Junsun technologies Co., Taipei, Taiwan) was used to deposit chromium and gold. The advantages of deposit are excellent coating adhesion and easily tunable microstructure [26], as well as fast deposition rates and high throughput per pass. Due to the insufficient adhesion of the gold vapor deposited on the PI film, it is easy to cause the gold to fall off. Therefore, chromium is used as an adhesion layer to reduce gold falling off.(c)The patterns of micro hydrogen, temperature, voltage, current, flow, and humidity sensors were defined using photolithography. Photolithography is a process that takes advantage of the decomposition of the photosensitive part of the positive photoresist, and this study uses AZ^®^ P4620 positive photoresist. First, a spin coater is used to apply the positive photoresist evenly to the substrate. It is then placed on a mask aligner and exposure system (AG-200-4N-D-SM, M&R Nano Technology Co., Taoyuan, Taiwan) with a flexible six-in-one micro-sensor photomask for exposure. The light-transmitting part of the photomask allows UV light to pass through the machine, while the part of the positive photoresist exposed to UV light will be removed. After three minutes of development, the photolithography process was completed, and a complete pattern was obtained. The developer used in this experiment is AZ^®^ 400K.(d)Wet etching was conducted with AG-835 gold etchant and Cr-7T chromium etchant to remove the metal not covered by photoresist.(e)A photoresist was reapplied to define the pattern of the hydrogen microsensor, the tin dioxide and platinum were sputtered on the hydrogen microsensor, and the photomask was removed.(f)LTC 9320 was applied as a protective coating. Then, the sensing areas and pins of the voltage, current, and humidity microsensors were redefined using the photolithography process to leave them exposed and not covered by a protective layer.(g)Finally, a coat of LTC 9305 was applied as the humidity sensing film of the humidity microsensor to complete the manufacturing process of the flexible six-in-one microsensor.

## 4. Calibration of Flexible Six-in-One Microsensor

Figure 3 shows an optical microscope image of the flexible six-in-one microsensor. The following introduces the calibration of hydrogen, temperature, humidity, flow, voltage, and current microsensors.

### 4.1. Hydrogen Calibration of Flexible Six-in-One Microsensor

First, this research embedded the hydrogen microsensor in the battery and used an eight-channel fuel cell testing machine to introduce oxygen at a constant temperature and flow, which allowed the surface of the hydrogen microsensor to adsorb the oxygen ions and resulted in increased resistance. Then, a constant temperature and constant speed of hydrogen were introduced, and the hydrogen took the oxygen ions on the surface of the hydrogen microsensor, which resulted in a drop in resistance. Hydrogen detection was performed using the resistance difference between the two. Hydrogen microsensors with different SnO_2_ film thicknesses will have different sensitivities, where the smaller the film thickness, the higher the sensitivity. Figure 4 shows the change of resistance of the hydrogen microsensor when oxygen and hydrogen were introduced at 30 °C (303.15 K). It can be clearly seen that the resistance dropped rapidly at the moment hydrogen was introduced, and then increased when oxygen was introduced. In addition, the different surface roughness will also affect its sensitivity, where the higher the roughness, the larger the detection surface area of the hydrogen microsensor. However, when a hydrogen microsensor is baked at a high temperature, it will reduce the surface roughness and sensitivity. Therefore, the fabrication process of the hydrogen microsensor must be conducted after the protective layer and the dielectric layer were baked (both the protective layer and dielectric layer must be baked at a high temperature of 240 °C (513.15 K) for 3 h). The surface roughness changed significantly before and after baking. Figure 5 shows the difference in sensitivity of the same microsensor before and after high-temperature baking.

### 4.2. Temperature Calibration of Flexible Six-in-One Microsensor

This study used the DENG YNG DS45 Drying Oven, which is a constant temperature oven for temperature correction and placed the flexible six-in-one microsensor and the BM-525 BRYMEN multimeter thermometer (Shining, Hsinchu, Taiwan) into the oven. After the reference control temperature setting was stable, it captured the resistance value of the temperature microsensor, resistance values were taken at every 10 °C increase in the operating temperature range, and the temperature microsensor was calibrated three times to obtain the average value of the calibration curve. Figure 6 shows the average calibration curves for six temperature microsensors, which shows that all the curves are almost linear.

### 4.3. Humidity Calibration of Flexible Six-in-One Microsensor

The sensing method of the humidity microsensor was conducted through capacitive detection, as it is less affected by temperature. However, as the change of the capacitance value was not large in the experiment, the humidity change could not be accurately measured. Finally, resistive detection was adopted for calibration under different temperature conditions, as resistance detection is affected by temperature.

Humidity correction was conducted in a constant temperature and humidity testing machine, and the relative humidity setting range was between 40% and 100%. The humidity was corrected at every 10% interval, and the temperature was fixed. Each time the relative humidity was raised, the micro-humidity sensor’s heater was reheated to evaporate any remaining moisture from the previous time. After each round became stable for 120 min, the NI PXI 2575 data collector was used to capture the resistance value of the humidity microsensor in real time, in order to obtain its calibration curve. Figure 7 shows the average calibration curve of the humidity microsensor at different temperatures; the humidity changed more at higher temperatures.

### 4.4. Flow Calibration of Flexible Six-in-One Microsensor

The flow microsensor was mainly placed on the oxygen side because only this part allows fluid to flow continuously. As the hydrogen side in the proton battery stack introduced water during charging, the oxygen side was introduced during discharge. Therefore, the flow microsensor must be calibrated with water and gas, individually. First, the flow microsensor was embedded inside the battery to avoid the vibration of the micro flow sensor due to the fluid, which will generate noise. Then, the power supply, flow microsensor, and NI PXI 2575 were connected in series to measure the current. As the power supply provides a regulated voltage to the flow microsensor, which generates heat, it will cause the resistance to rise and the current to drop. Then, as the fluid carries away the generated heat, the resistance will drop and the current will rise, and this was used to measure the flow.

The oxygen used in this study was provided by an eight-channel fuel cell testing machine. The oxygen calibration range was based on 400 mL/min (6.67 × 10^−6^ m^3^/s), and every 100 mL/min (1.67 × 10^−6^ m^3^/s) increased the interval to 900 mL/min (1.5 × 10^−5^ m^3^/s). The water flow was provided by a YOTEC WS Series variable speed flow pump, where the water flow correction range was based on 100 mL/min (1.67 × 10^−6^ m^3^/s), and every 10 mL/min (1.67 × 10^−7^ m^3^/s) increased the interval to measure 150 mL (1.5 × 10^−4^ m^3^). The calibration results are shown in Figure 8 and Figure 9.

### 4.5. Voltage and Current Calibration of Flexible Six-in-One Microsensor

This study used a multimeter to confirm whether the measurements of the voltage microsensor and current microsensor were correct, and this calibration procedure is intended to confirm the battery voltage and current value. After confirmation, the microsensor head was connected to the battery, as shown in Figure 10.

### 4.6. Accuracy of Calibration Machine

Despite the care that was taken while calibrating, there may still be machine errors. The error of the calibration machine causes the error of the microsensors. Therefore, we explain the accuracy of the machine. We used an eight-channel fuel cell testing machine to set up a calibration environment for hydrogen and flow microsensors, and the accuracy of this machine was 2.5%. We use DENG YNG DS45 Drying Oven to set up the calibration environment of temperature microsensor, and the accuracy of this machine is ±0.5 °C. We use constant temperature and humidity testing machine to set up the calibration environment of humidity microsensor, and the accuracy of this machine is 0.1 °C.

## 5. Internal Measurements of Proton Battery Stack

This study embedded the developed flexible six-in-one microsensor into a proton battery stack (Figure 11a) for internal measurements. The flexible six-in-one microsensors were placed upstream and downstream of the oxygen side and upstream of the hydrogen side, as shown in Figure 11b,c.

### 5.1. Internal Voltage and Current Measurement during Proton Stack Charging

The internal voltage was measured by connecting the voltage microsensors at the oxygen side and the hydrogen side, and the measurement results are shown in Figure 12. Since the two sides of the proton exchange membrane were charged separately by the power source, they were slightly different. A comparison of the value of the power supply showed that the measured voltage was similar, and the upward trend was the same, thus, it was inferred that the proton battery stack was in a stable electrochemical reaction state at this time. 

As the proton battery stack in this experiment gave a stable constant current when charging, and the measured internal current was also in the same stable state, it was inferred that there was a stable electrochemical reaction inside. Current density variation is not more than plus or minus 0.3 mA/cm^2^ (3 A/m^2^), as shown in Figure 12.

### 5.2. Internal Temperature Measurement during Proton Battery Stack Charging

Figure 13 shows the internal temperature measurements for the oxygen side and hydrogen side during charging, respectively. It can be seen that the temperature of the water outlet upstream of the oxygen side B was higher, which is because the heat generated during charging is the first outlet after the reaction. Compared to other oxygen end drains, the oxygen side B heat has not yet escaped due to water circulation, so the temperature is higher. As the sulfuric acid on the hydrogen side was not in the flow exchange with external sulfuric acid, its temperature did not drop due to the contact with the fluid in the environment. The heat caused by the charging reaction cannot be discharged. Thus, the temperature on the hydrogen side increased slightly.

### 5.3. Internal Flow Measurement during Proton Battery Stack Charging

The internal flow of the proton battery stack is shown in Figure 14. As there is a water inlet under oxygen side A, its flow rate was relatively unstable; while the subsequent inlets and outlets were relatively stable, the flow rate slowed down due to the blockage in the carbon felt.

### 5.4. Internal Humidity Measurement during Proton Battery Stack Charging

The humidity inside the proton battery stack is shown in Figure 15. This value remained around 100% because the oxygen side entered the water when the proton battery was charged.

### 5.5. Internal Hydrogen Measurement during Proton Battery Stack Charging

Figure 16 shows the hydrogen sensing inside the proton battery stack. As the internal environment was full of sulfuric acid, the hydrogen microsensor was not exposed to oxygen to return to its pre-measurement state. Therefore, as seen in the picture, it can be measured only when the hydrogen gas started to appear 20 min after starting to charge, meaning that when the activated carbon on the hydrogen side was almost saturated, the hydrogen started to bubble.

## 6. Improvement of Flexible Six-in-One Microsensor

During the experiments, it was found that the flexible six-in-one microsensors using ceramic circuit boards are prone to bending when embedded in the battery stack due to the weight of the ceramic circuit board itself and the weight of the conductive wires. This bending will cause the double-sided adhesive to be dislodged from the flexible six-in-one microsensor, which will cause the conductive gel and silver paint pen to peel off and damage the sensor. There is a possibility that the split created when cutting the flexible six-in-one microsensor may cause it to break directly along the split when it is bent. Therefore, a new connection method for the flexible six-in-one microsensor was found to solve this problem.

The length of the flexible six-in-one microsensor cannot be changed due to the limited manufacturing machine. This makes it impossible to place the wires on the platform when connecting to the ceramic circuit board, which causes the overall impact of gravity. In order to maintain the bendability of the flexible six-in-one microsensor, a flexible board was used to replace the ceramic board. To make up for the lack of overall length, the flexible board can be directly connected to the flexible board adaptor. The flexible board is connected to the flexible six-in-one microsensor using anisotropic conductive adhesive (as Figure 17). It has the characteristics of vertical conductivity and horizontal insulation, good adhesion, and thin thickness, so it is used as the adhesive material. After testing, the flexible six-in-one microsensors were successfully connected to the flexible board, and problems such as peeling and breaking were improved.

## 7. Conclusions

This study successfully developed a flexible six-in-one microsensor using MEMS technology. This flexible six-in-one microsensor offers many advantages, including six sensing capabilities, resistance to electrochemical environments, real-time measurements, and being able to be placed anywhere within a proton battery stack, among others. The thin thickness allows the microsensor to be embedded in the proton cell stack with a good seal to avoid liquid leakage. In the process, the manufacturing steps were adjusted, and it was confirmed that the sensitivity of the hydrogen microsensor would change as a result of the process steps. Moreover, connecting the proton batteries in parallel can improve battery life. The wide variation in hydrogen absorption rates and release rates exhibited by proton cell stacks is probably due to the temperature and the fact that the proton exchange membrane used is for proton exchange membrane water electrolysis. The ratio of materials in the catalytic layer is different from that of the catalytic layer of the proton exchange membrane used in fuel cells. This results in unsatisfactory performance in the discharge process.

The sensor has fallen off and broke during the experiment. We succeeded in finding a new circuit solution for a flexible six-in-one microsensor. A flexible board is used for the connection, and anisotropic conductive adhesive is used for the adhesion. The advantage of bendability is maintained.

## Figures and Tables

**Figure 1 membranes-12-00779-f001:**
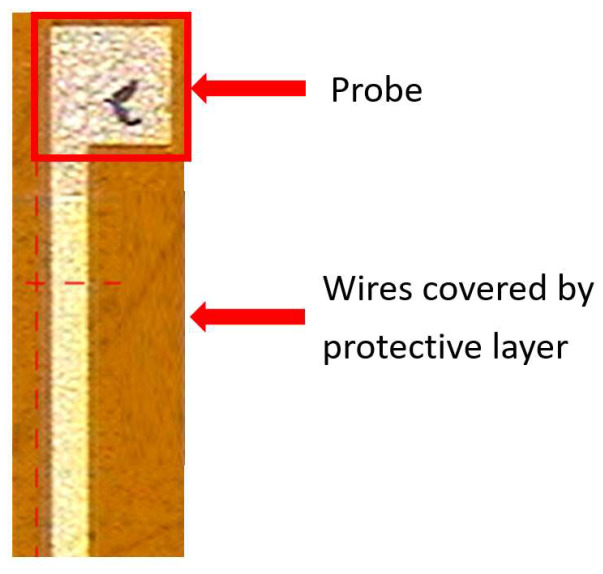
Schematic diagram of voltage and current microsensor.

**Figure 2 membranes-12-00779-f002:**
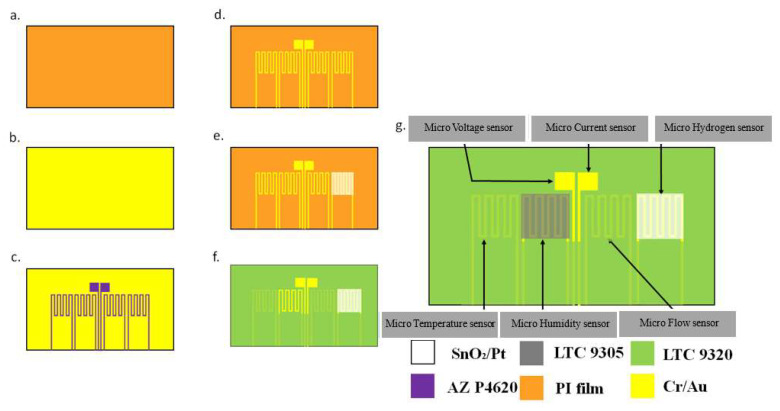
Process flowchart of flexible six-in-one microsensor.

**Figure 3 membranes-12-00779-f003:**
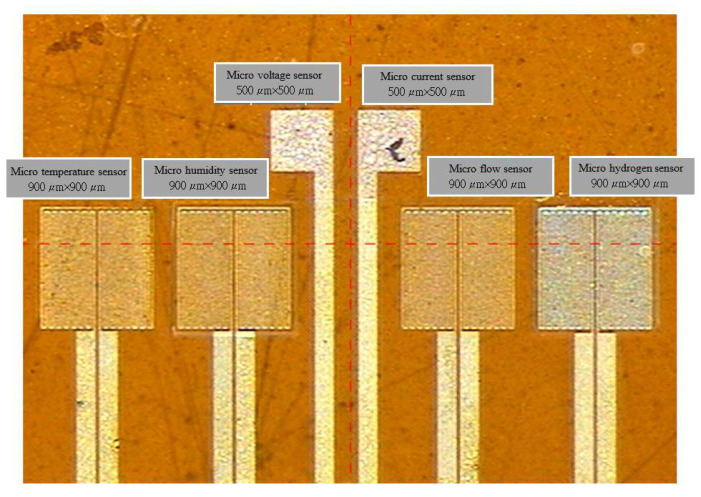
Optical microscope diagram of flexible six-in-one microsensor.

**Figure 4 membranes-12-00779-f004:**
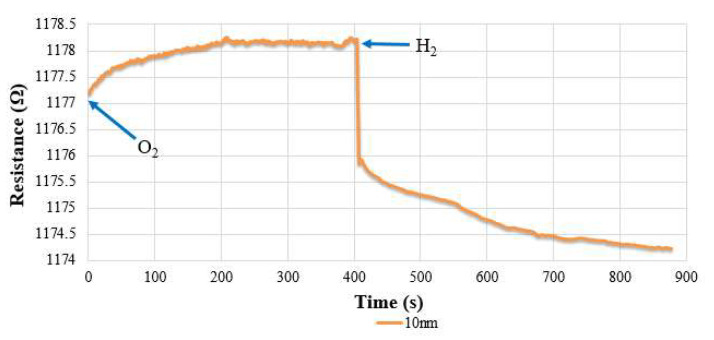
The resistance change of the hydrogen microsensor with hydrogen and oxygen.

**Figure 5 membranes-12-00779-f005:**
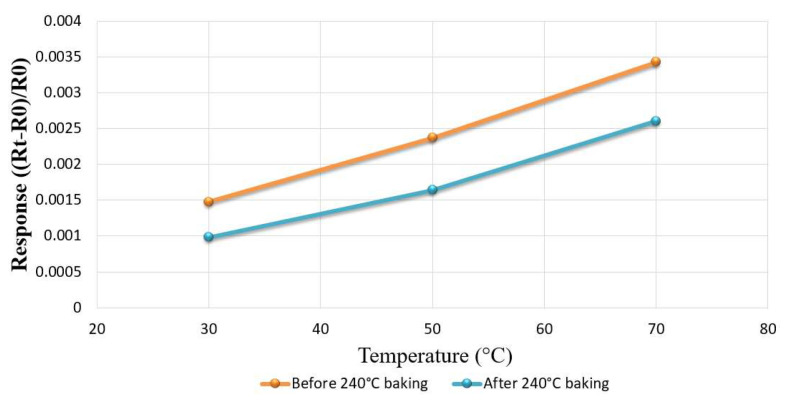
Comparison of the surface sensitivity of the hydrogen microsensor before and after baking.

**Figure 6 membranes-12-00779-f006:**
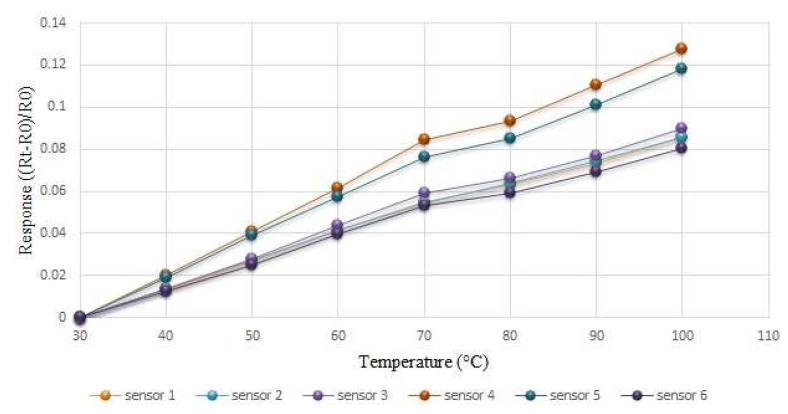
Average calibration curve of six temperature microsensors.

**Figure 7 membranes-12-00779-f007:**
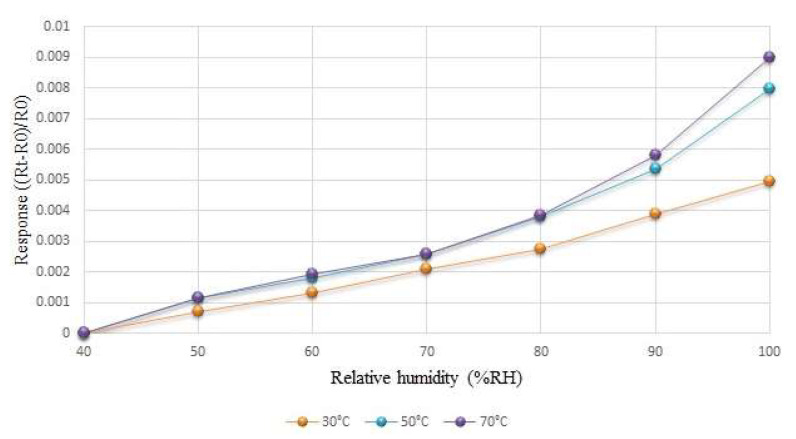
The average calibration curve of the humidity microsensor at different temperatures.

**Figure 8 membranes-12-00779-f008:**
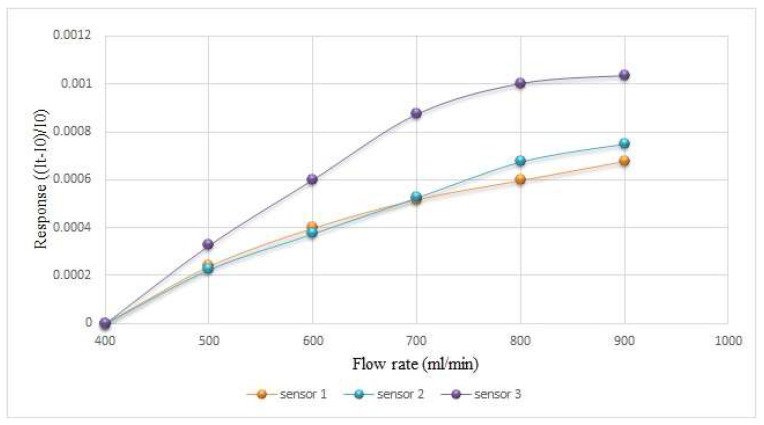
Calibration of micro flow sensor in oxygen state.

**Figure 9 membranes-12-00779-f009:**
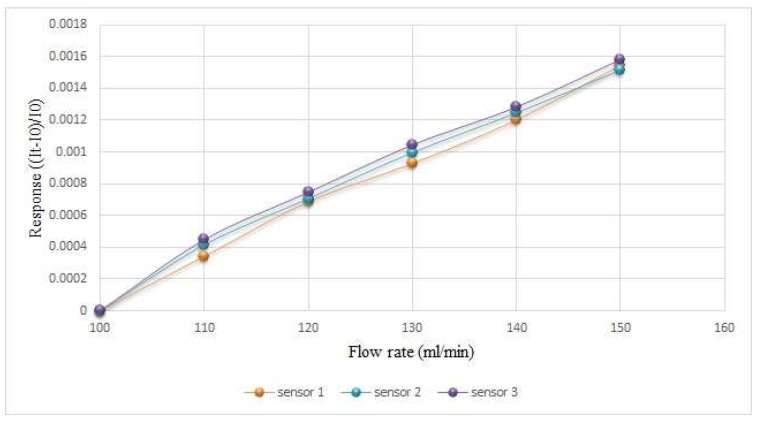
Calibration of micro flow sensor in water-passing state.

**Figure 10 membranes-12-00779-f010:**
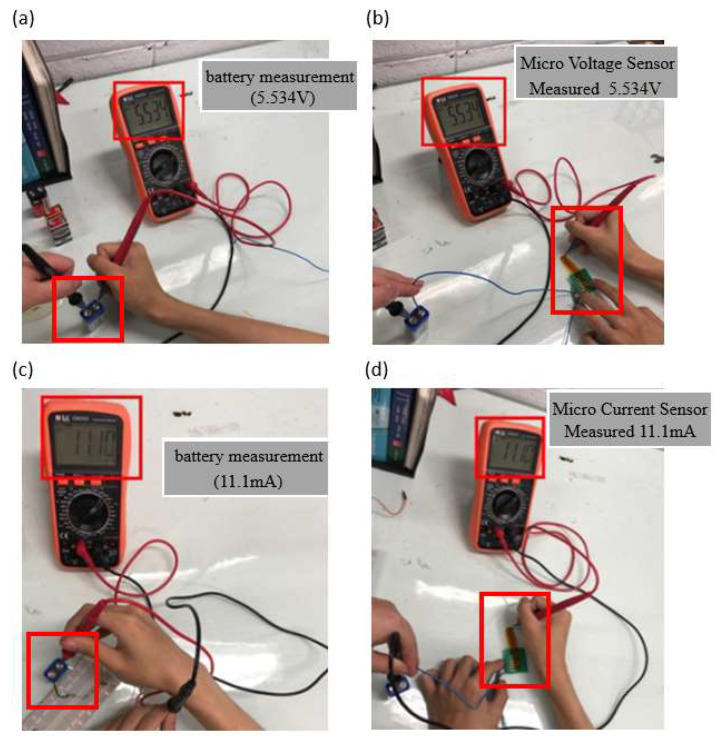
(**a**) Battery voltage measurement; (**b**) micro voltage sensor measurement of battery; (**c**) battery current measurement; (**d**) micro current sensor measurement of battery.

**Figure 11 membranes-12-00779-f011:**
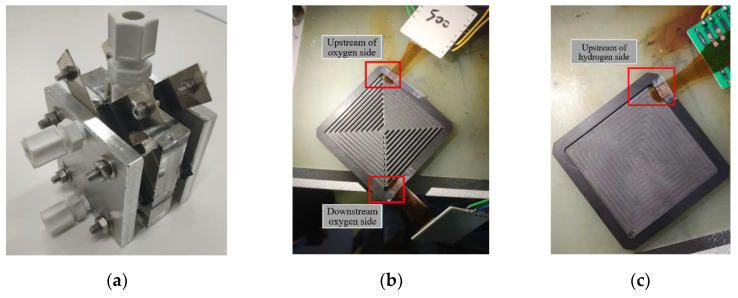
(**a**) Entity of a proton battery stack; (**b**) the position of the flexible six-in-one microsensor embedded in the oxygen side; (**c**) the position of the flexible six-in-one microsensor embedded in the hydrogen side.

**Figure 12 membranes-12-00779-f012:**
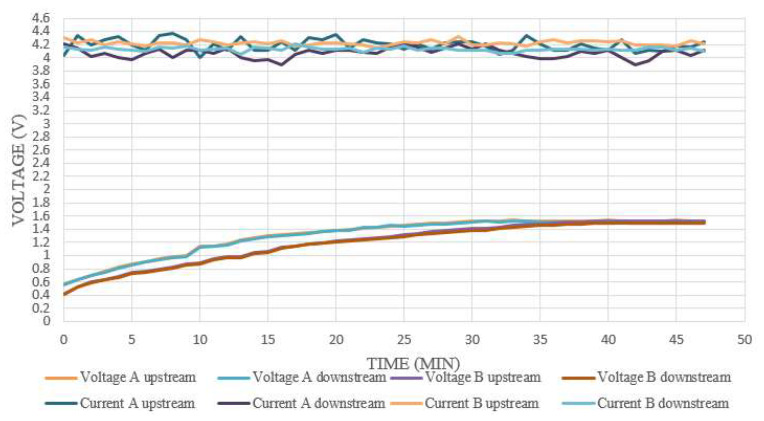
Internal voltage and current measurement during proton battery stack charging.

**Figure 13 membranes-12-00779-f013:**
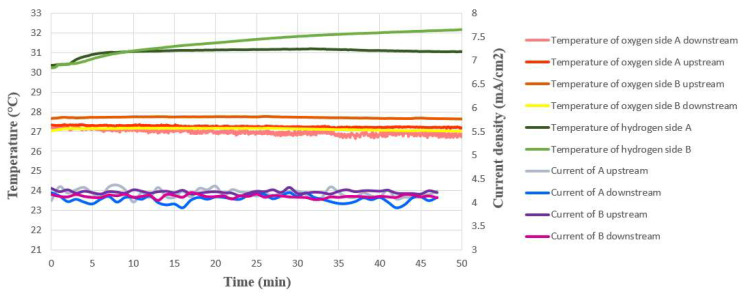
Internal current and temperature measurement during proton battery stack charging.

**Figure 14 membranes-12-00779-f014:**
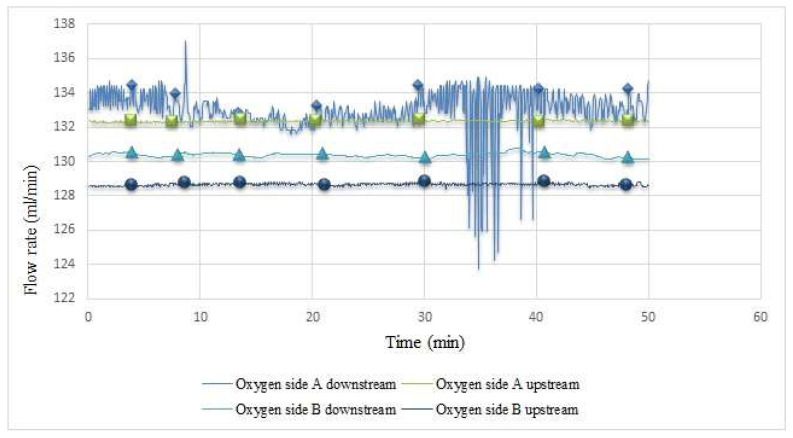
Internal flow measurement during proton battery stack charging.

**Figure 15 membranes-12-00779-f015:**
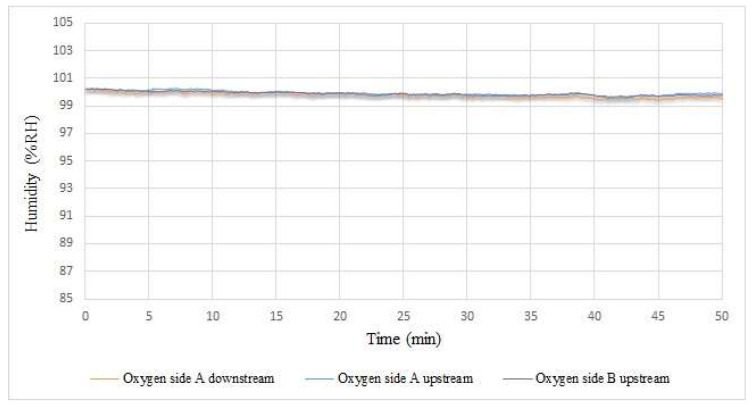
Internal humidity measurement during proton battery stack charging.

**Figure 16 membranes-12-00779-f016:**
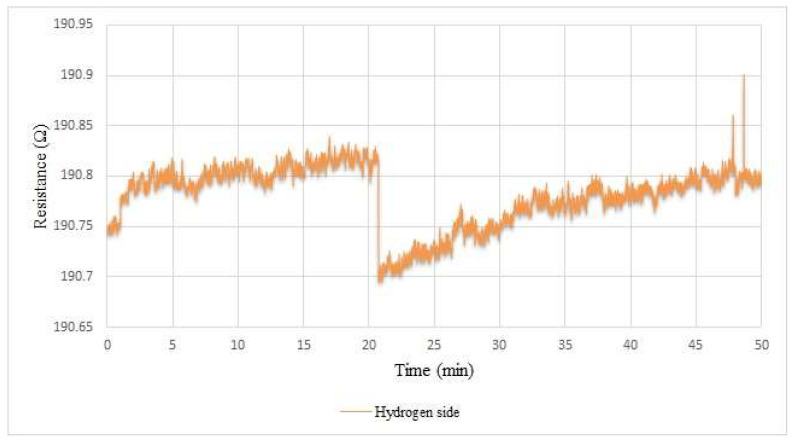
Internal hydrogen measurement during proton battery stack charging.

**Figure 17 membranes-12-00779-f017:**
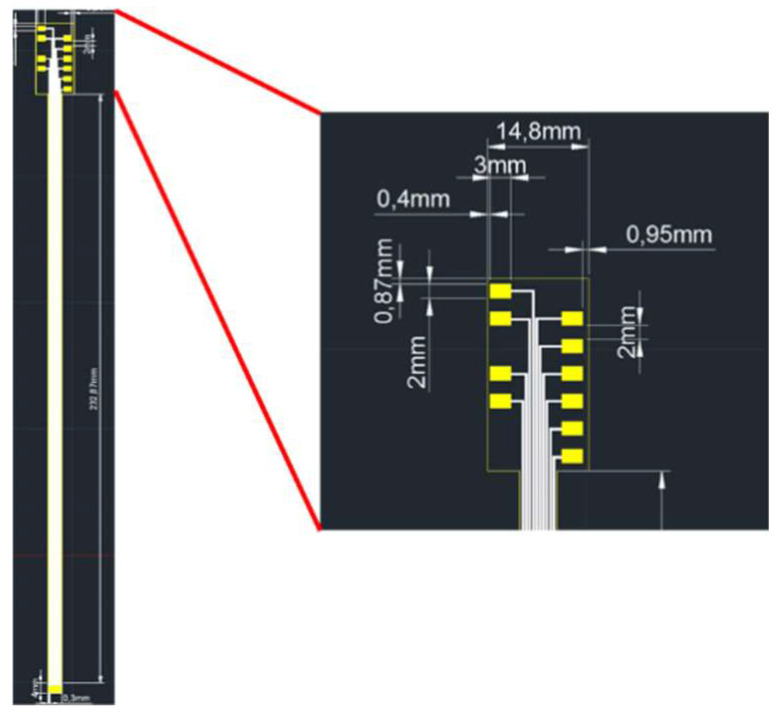
Flexible board design.

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
