# Peer review of "A Proton Battery Stack Real-Time Monitor with a Flexible Six-in-One Microsensor"

_membranes, 2022, doi:10.3390/membranes12080779_

Round 1

Reviewer 1 Report

The document is considered ready to be published.

Reviewer 2 Report

The manuscript was revised according to the journal rules.

Few revisions are required and they are reported below:

- please move the nomenclature section at the beginning pf the paper

- I suggest to add all acronyms, parameters with SI as unit of measure

- I suggest to add one or two sentences on the H2 leaks risks, you could consider HYTUNNEL project from EU.

- I suggest to add references for the H2 porous material

- figure 1 should be revised maybe with real elements

- figures 4,5 and 7 should be revised, they seem blurry

- Since the work presented is experimental in nature, I recommend adding the uncertainty bars in the measurements taken, perhaps it might be plausible to include a section on the uncertainty analysis of the instruments used.

- figures presented should be reduced, maybe could be used the Supplementary material section

Author Response

This manuscript is a resubmission of an earlier submission. The following is a list of the peer review reports and author responses from that submission.

Round 1

Reviewer 1 Report

The traditional semiconductor sensor usually consists of two pairs of electrodes, in which one pair is the working electrode and another pair is used as a heating electrode. In the present work, the electrode configuration of the sensing chip is not clear. Please proved a photo of the sensing chip and experimental setup, so that I can understand how the sensing temperature was controlled.

The sensors displayed high selectivity towards H2, but the mechanism behind the high selectivity is not well understood. Please try to explain it in a more deep way. In addition, the response toward other types of volatile gas needs to be measured.

The sensing mechanism is nowhere mentioned. Please go into detail into the investigation of the active site of the sensing materials.

The authors need to be checked the grammatical and typographical errors throughout the manuscript. Also, authors must use uniform abbreviations for the proposed sensing material and current collector, etc.

Reviewer 2 Report

The manuscript examines an internal diagnostic tool for a proton battery cell using microelectromechanical systems (MEMS) technology and the study of the innovatively developed hydrogen microsensor and integrated microsensors for voltage, current, temperature, humidity, and flow. , as previously developed by our laboratory, to complete the six-in-one flexible microsensor. However, in general, it is considered that the document presented does not contain the necessary information for its publication in Membranas.
There are many syntax errors, the English should be improved.
The document is considered to have little originality.
The organization of the entire manuscript is not adequate, as it is confusing to read and understand.
The document presents too many Figures (22) without proper discussion and supported by references.
It is felt that many of the graphs can be put together and discussed better, for example Figure 17, Figure 18 and Figure 19 can be put together, with double y-axis.
It is necessary to carry out a broader bibliographic review to support the work.
In conclusion, I consider that the manuscript should not be published in the journal. I suggest a new shipment with a vast improvement.

Reviewer 3 Report

The article was revised according to the journal's specifications. The topic of the manuscript is interesting and deserves to be investigated. 
Some revisions are listed below:
- I would suggest revising the abstract, it should be more focused on the concrete results obtained
- I would add a section on nomenclature, in which parameters, acronyms and units could be included
- the introductory section could be enlarged by using more references identified in the literature
- the materials and methods section could be improved by adding details of the instruments used 
- Figures in the results section should be revised.
- a section on the uncertainty analysis of the results achieved should be added
- in the results section I would expand the discussion to include published studies presented in the literature.
- the conclusion section should be expanded and deepened.

Round 2

Reviewer 2 Report

The manuscript examines an internal diagnostic tool for a proton battery cell using microelectromechanical systems (MEMS) technology and the study of the innovatively developed hydrogen microsensor and integrated microsensors for voltage, current,temperature, humidity, and flow, as previously developed by our laboratory, to complete the six-in-one flexible microsensor. However, in general, it is considered that the document presented does not contain the necessary information for its publication in Membranes

1. The authors must carry out an extensive revision in the organization of the document, since several Figures are repeated, for example, Figure 17 and  Figure 18.

2. In section 4.4. Voltage and current calibration of flexible 6-in-1 six-in-one microsensor, it is necessary to unite Figure 12 and Figure 13, with a respective analysis that can be done by using arrows, circles, in general, tools that may work to better explain the Figure.

3. In section 5. Calibration of flexible 6-in-1 six-in-one microsensor, it is necessary to join Figure 14, Figure 15 and Figure 16 with the respective items a), b) and c).

4. In order to better organize the document, it is considered that the sections can be joined, for example:

5.1. Internal voltage measurement during proton stack charging

5.2. Internal current measurement during proton battery stack charging

The above can join:

5.1. Internal voltage and current measurement during proton stack charging

Due to the above, it would be possible to unite both Figures 17 in the same one.

5. It is considered that many of the graphs can be put together and discussed better, for example, it is possible to join a single Figure "X", Figure 18, 19 and 20. Added, Figure 18 is repeated. The entire document can be improved with changes such as those already mentioned.

6. It is necessary to carry out a broader bibliographic review to support the work. The authors should review the following references to ensure a better discussion of the results based on MEMS technology and systems for energy conversion in general.

In conclusion, I consider that the manuscript should not be published in the journal. I suggest a new shipment with a vast improvement

Reviewer 3 Report

The revision was accomplished even if there are few revisions to be accomplished:

  • the nomenclature section is missing
  • point 5 should be followed
